# A Causal Lens for Learning Long-term Fair Policies

**Jacob Lear & Lu Zhang**
Department of Electrical Engineering and Computer Science
University of Arkansas
{jdlear,lz006}@uark.edu

## Abstract

Fairness-aware learning studies the development of algorithms that avoid discriminatory decision outcomes despite biased training data. While most studies have concentrated on immediate bias in static contexts, this paper highlights the importance of investigating long-term fairness in dynamic decision-making systems while simultaneously considering instantaneous fairness requirements. In the context of reinforcement learning, we propose a general framework where long-term fairness is measured by the difference in the average expected qualification gain that individuals from different groups could obtain. Then, through a causal lens, we decompose this metric into three components that represent the direct impact, the delayed impact, as well as the spurious effect the policy has on the qualification gain. We analyze the intrinsic connection between these components and an emerging fairness notion called benefit fairness that aims to control the equity of outcomes in decision-making. Finally, we develop a simple yet effective approach for balancing various fairness notions.

## 1 Introduction

Artificial intelligence and machine learning decision-making systems are being increasingly implemented in real-world scenarios Zhang et al. (2017a); Johnson et al. (2016); Baker & Hawn (2021); Lee & Floridi (2021); Schumann et al. (2020); Berk et al. (2021). Real-world data, influenced by social and historical contexts, often carries biases related to gender, race, and other factors. These biases can be inadvertently embedded in the algorithms, leading to discriminatory outcomes. As a result, fairness-aware learning, which aims to satisfy various fairness constraints alongside the usual performance criteria in machine learning, has received increasing attention. A body of literature on fairness-aware learning focuses on developing fair policies in reinforcement learning (RL) Sutton & Barto (2018). For a comprehensive survey on fair RL, please refer to Gajane et al. (2022).

Most studies in fairness-aware learning focus only on the immediate implications of bias in a static context. These works typically quantify the fairness of model predictions or outcomes in a static population. However, real decision-making systems usually operate dynamically, and the decisions made by these systems have long-term consequences. The literature has shown that fairness notions and techniques focusing on the immediate bias produced by automated decisions cannot guarantee to protect disadvantaged groups in the long run Liu et al. (2018). It has hence been proposed to consider the delayed impact of automated decisions in sequential decision-making systems due to the interplay between the decisions and individuals' reactions Liu et al. (2020). For example, Zhang et al. (2020) proposes to use the equilibrium of the dynamics of population qualifications across different groups of individuals as a measure of the delayed impact of decisions on fairness. In this paper, when the context is clear, we refer to the instantaneous fairness concerns, such as demographic parity Feldman et al. (2015), equal opportunity Hardt et al. (2016), and causality-based notions like direct and indirect discrimination Zhang et al. (2017b), as *short-term fairness*. On the other hand, we refer to fairness concerns that arise over time due to dynamic user-decision interactions as *long-term fairness*, which is the primary focus of this paper.

Given that the objective of RL is typically to maximize the long-term (discounted) reward, long-term fairness has been studied in the context of RL where Markov Decision Processes (MDPs) are

utilized to model and learn the system dynamics Jabbari et al. (2017); Ge et al. (2021); Wen et al. (2021); Yu et al. (2022); Hu et al. (2023); Yin et al. (2023). However, such requirements may be conflict with long-term fairness constraints. For instance, Hu et al. (2024) models the sequential decision-making system with a temporal causal graph and captures long-term and short-term fairness as causal effects transmitted through different paths in the temporal causal graph. The authors demonstrated that achieving long-term fairness could be impeded by the requirement of sensitive attribute unconsciousness, i.e., no direct causal effect of the sensitive attribute on the model decision, implying an inherent conflict between long-term and short-term fairness. Thus, understanding the connection between long-term and short-term fairness is critical for developing comprehensive strategies that balance both fairness requirements.

In this work, we study the specific task of analyzing the long-term fairness that can be achieved in the context of RL. We focus on the impact of the model's decisions on individuals' qualifications and aim to understand how long-term fairness intertwines with immediate short-term fairness concerns during sequential decision-making. To this end, we propose a novel decomposition of long-term fairness that differs from existing causal decomposition techniques applied to the advantage function, such as Pan & Schölkopf (2024). We begin by introducing a general framework for studying long-term fairness in RL, where we assume a flexible qualification gain function that measures an individual's qualification gain as they transition between qualification states. Accordingly, we define the expected total qualification gain accumulated from a given state while following a specific policy as a state value function, which can be expressed using Bellman's equation. Based on that, long-term fairness can be readily formulated based on the disparity in the expectation of the state value function across different groups. We then conduct a causal decomposition of the qualification gain disparity to identify the various sources of inequality, noting that the qualification gain is influenced by both the policy and the environment. We end up obtaining three components of the qualification gain disparity: (1) the Direct Policy Effect (DPE) which represents the direct causal effect of the policy on the long-term qualification gain; (2) the Indirect Policy Effect (IPE) which represents the indirect causal effect of the policy on the long-term qualification gain through the environment; and (3) the Spurious Policy Effect (SPE) which represents the spurious effect only due to the environment. Interestingly, we identify an inherent connection between these components and an emerging fairness notion called benefit fairness that aims to control the equality of the outcome of the decision-making Plecko & Bareinboim (2023). Our analysis shows that benefit fairness may not necessarily conflict with long-term fairness, suggesting that it is possible to reconcile short-term and long-term fairness objectives. This connection may offer insights for designing decision-making systems where the long-term and short-term objectives are aligned. Finally, we provide a simple yet effective approach to strike a balance between qualification gain parity and benefit fairness.

## 2 RELATED WORK

Early research in long-term fair machine learning focuses on specific applications and scenarios. For example, the authors in Holzer (2007) examine the long-term effects of affirmative action in hiring and highlight its positive impacts on minority and low-income communities. The authors in Hu & Chen (2018) study long-term fairness in a two-stage labor market and construct a dynamic reputational model that shows how unequal access to resources leads to different investment choices, which in turn reinforces unequal outcomes between groups. The study in Liu et al. (2018) investigates the delayed impact of decisions in lending scenarios using a one-step feedback model and reveals that short-term fairness notions generally do not reshape the population to foster long-term improvements.

Reinforcement learning (RL) offers a solution for modeling the long-term impact of decisions through Markov decision processes. Fairness methods in RL designed to address long-term effects have been proposed. The study in Jabbari et al. (2017) presents a fairness constraint that guarantees an algorithm will not favor one action over another if the latter has a higher long-term reward. However, this notion of fairness is solely based on the reward of each action and does not consider demographic information. The research in Ge et al. (2021) studies long-term fairness and formulates a Constrained Markov Decision Process (CMDP) for recommendation systems. In Wen et al. (2021), fairness constraints are defined based on the average rewards received by individuals across different groups, and two algorithms have been designed to learn policies that meet these fairness constraints. Yu et al. (2022) suggests incorporating fairness requirements into policy optimization by regulariz-

ing the advantage assessment of different actions. Similarly, Hu et al. (2023) incorporates long-term fairness constraints into the advantage function but proposes a pre-processing technique called action massaging to address short-term fairness, aiming to balance both requirements. The research in Yin et al. (2023) also demonstrates that achieving long-term fairness may require sacrificing short-term incentives, It also develops probabilistic bounds on cumulative loss and cumulative fairness violations. Finally, different from the above work, Henzinger et al. (2023) develops a monitor that continuously tracks events generated by the system in real-time, providing an ongoing assessment of the system's fairness with each event.

Another related research direction is performative prediction and performative optimization which have emerged as powerful tools for addressing distributional shifts that occur when deployed models influence the data-generating process Perdomo et al. (2020); Jia et al. (2024). This line of research also offers valuable insights into achieving long-term fairness in dynamic systems. For instance, Jin et al. (2024) examines issues of polarization and unfairness within performative prediction settings. They demonstrate that traditional penalty terms for long-term fairness may fail in performative prediction settings and propose new fairness mechanisms that can be incorporated into iterative algorithms, such as repeated risk minimization, which has been shown to be effective in these settings.

Our research complements related studies by examining the trade-off between long-term and short-term fairness in RL settings from a causal perspective and establishing a connection between them.

## 3 METHODOLOGY

### 3.1 PRELIMINARIES

We adopt Pearl's structural causal model (SCM) and causal graph framework Pearl (2009) to facilitate modeling the Markov decision process and formulating long-term fairness. Here, we provide a brief overview of the fundamentals of SCM. A detailed introduction to SCM can be found in Pearl (2010). An SCM is a mathematical framework used in causal inference to represent and analyze the relationships between variables in a system. It provides a formal way to describe how changes in one variable can causally affect other variables, enabling the analysis of cause-and-effect relationships. An SCM defines the causal dynamics of a system through a collection of structural equations. Each SCM is associated with a causal graph that includes a set of nodes to represent variables and a set of directed edges to depict direct causal relationships.

Causal inference within the SCM is enabled through interventions, as described in Pearl (2009). A hard intervention assigns specific constant values to certain variables, while a soft intervention establishes a functional relationship for some variables in response to others Correa & Bareinboim (2020). A causal path from node $X$ to node $Y$ in a causal graph is a directed sequence of arrows leading from $X$ to $Y$. The total causal effect refers to the impact of $X$ on $Y$ when the intervention is transmitted along all causal paths connecting $X$ to $Y$. If the intervention is restricted to only a subset of these causal paths, the resulting effect is known as the path-specific effect Avin et al. (2005).

### 3.2 FORMULATING LONG-TERM FAIRNESS

Consider a discrete-time sequential decision-making process applied to a certain population, described as a Markov decision process (MDP). Each individual in the population is described by a sensitive feature and a qualification state. The sensitive feature $S$ is assumed to be a binary variable in this paper for ease of representation, i.e., $S \in \{s^+, s^-\}$. The qualification state, denoted as $\mathbf{X}$, represents the individual's suitability or potential performance in the given context relevant to a specific decision-making task. It could be a variable vector and may evolve over time, reflecting the dynamic nature of an individual's attributes or skills. At each time step, a Markov policy $\pi(d|\mathbf{x}, s)$ is used to make decisions $D \in \{d^0, d^1\}$ where $d^0$ represents negative decision or non-treatment, and $d^1$ represents positive decision or treatment. A reward $R$ is received following each decision and the utility of the policy is defined as the expected total reward received during the process. Meanwhile, individuals may also take actions upon receiving the decisions, which may change their qualification states at the next time step. This interplay is captured by a transition probability $P(\mathbf{x}_{t+1}|\mathbf{x}_t, d_t, s)$.

The above process can be described by the causal graph shown in Fig. 1 which can represent either a randomly sampled individual or a population undergoing the decision cycles. In general, at each

time step $t$, the agent selects an action $d_t$ based on the policy $\pi$. This causes the qualification state to transition from $\mathbf{x}_t$ to an intermediate state $(\mathbf{x}_t, d_t)$, which subsequently transitions to the next state $\mathbf{x}_{t+1}$ according to the transition probability. Without loss of generality, we assume that the sensitive feature $S$ may influence any part of the process, including the initial state distribution, the transition probability, the policy function, etc.

A metric of long-term fairness typically considers the equilibrium of qualification states across different groups. It can also be treated as the causal effect of the sensitive feature on the qualification states. For instance, the authors in Hu et al. (2024) investigate long-term fairness by examining the path-specific effect, i.e., the causal effect transmitted through all paths from $s$ to $\mathbf{x}_T$ in the graph. They conclude that the objective of eliminating this causal effect may conflict with the requirement of sensitive attribute unconsciousness.

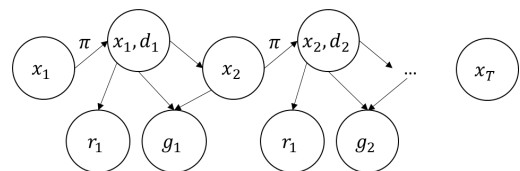

Figure 1: Causal graph for representing the discrete-time sequential decision-making process where $S$ is omitted. At each time step $t$, the state $\mathbf{x}_t$ transits to the intermediate state $\mathbf{x}_t, d_t$ based on the policy $\pi$, and then transits to $\mathbf{x}_{t+1}$ based on the transition probability. The reward $r_t$ is received based on the decision/action $\mathbf{x}_t$, and the qualification gain is obtained based on both $\mathbf{x}_t, d_t$ and the next state $\mathbf{x}_{t+1}$.

In this work, we explore long-term fairness and establish a general formulation within the framework of MDPs. The typical objective of MDPs is to maximize the expected cumulative reward for an agent interacting with an environment over time. Taking fairness into consideration, different from previous work that incorporates long-term fairness as an advantage regularization (e.g., Yu et al. (2022); Hu et al. (2023)), we argue for considering the expected total change in the qualification state when the individual interacts with the decision policy. To achieve this, we introduce a flexible *qualification gain function*, denoted as $g^s(\mathbf{x}, \mathbf{x}')$, which quantifies the increase/decrease in qualifications when an individual from the sensitive group $s$ transitions from state $\mathbf{x}$ to state $\mathbf{x}'$. This qualification gain function can be defined in any form as long as it satisfies the additive property over intervals, i.e., for all intermediate states $\mathbf{x}_1, \ldots, \mathbf{x}_n$ between $\mathbf{x}$ and $\mathbf{x}'$, we have $g^s(\mathbf{x}, \mathbf{x}') = g^s(\mathbf{x}, \mathbf{x}_1) + \cdots + g^s(\mathbf{x}_n, \mathbf{x}')$. Then, for a given trajectory in the sequential decision-making process, the total qualification gain from transitioning from the initial state to the final state is equivalent to the cumulative qualification gain throughout the process, i.e., $\sum_{t=0}^{T} g^s(\mathbf{x}_t, \mathbf{x}_{t+1})$.

To define the expected qualification gain an individual can achieve through the deployment of this policy, we employ the notations from Hu & Zhang (2022), where the policy deployment is treated as a soft intervention, denoted as $do(\pi)$. Meanwhile, in line with causality-based fairness notions, we consider the qualification gain individuals would achieve if their sensitive features were altered to different values. This involves performing hard interventions on the sensitive feature $S$. By conducting both interventions, we consider the interventional expected qualification gain, denoted as $V_{do(\pi,s)}(\mathbf{x})$. Adopting the causal perspective of reinforcement learning Zeng et al. (2023), the interventional expected qualification gain can be defined as the state value function $V_{do(\pi,s)}(\mathbf{x}) := \mathbb{E}_\pi \left[ \sum_{t=0}^{T} g^s(\mathbf{x}_t, \mathbf{x}_{t+1}) \mid \mathbf{x}_0 = \mathbf{x} \right]$. This state value function above captures the cumulative impact of decisions over time, enabling us to address the need for equity in decision-making systems over extended periods. A natural long-term fairness requirement, therefore, is to ensure that the average expected qualification gain is equal across different groups over time. Consequently, we define qualification gain parity as a general metric for long-term fairness, as detailed below.

**Definition 1 (Qualification Gain Parity)** *We say that a policy $\pi$ exhibits qualification gain parity if $C_\pi(\theta) = \mathbb{E}[V_{do(\pi,s+)}(\mathbf{x}_0)] - \mathbb{E}[V_{do(\pi,s-)}(\mathbf{x}_0)]$ is equal to zero.*

### 3.3 POLICY OPTIMIZATION WITH CONSTRAINTS

Qualification gain parity can be achieved through policy optimization with constraints. To this end, we first express the state value function using Bellman's equation as follows.

$$V_{do(\pi,s)}(\mathbf{x}) = \sum_d \pi(d|\mathbf{x}, s) Q_{do(\pi,s)}(\mathbf{x}, d), \tag{1}$$

where

$$Q_{do(\pi,s)}(\mathbf{x},d) = \sum_{\mathbf{x}'} P(\mathbf{x}'|\mathbf{x},d,s)\left(g^s(\mathbf{x},\mathbf{x}') + V_{do(\pi,s)}(\mathbf{x}')\right)$$

is the action value function. By following the Policy Gradient Theorem Sutton & Barto (2018), the gradient of the state value function Eq. (1) is given by $\nabla_\theta V_{do(\pi,s)}(\mathbf{x}) = \sum_{\mathbf{x}'} \eta^{\pi,s}(\mathbf{x} \to \mathbf{x}') \sum_d \nabla_\theta \pi(d|\mathbf{x}',s) Q_{do(\pi,s)}(\mathbf{x}',d)$ where $\eta^\pi(\mathbf{x} \to \mathbf{x}')$ is the probability of transitioning from state $\mathbf{x}$ to state $\mathbf{x}'$ with policy $\pi$ after an arbitrary number of steps. Thus, the gradient of $C_\pi(\theta)$ is given by $\nabla_\theta C_\pi(\theta) = \alpha\left(\mathbb{E}_{\pi|s^+}\left[\nabla_\theta \ln \pi(d|\mathbf{x},s^+) Q_{do(\pi,s^+)}(\mathbf{x},d)\right] - \mathbb{E}_{\pi|s^-}\left[\nabla_\theta \ln \pi(d|\mathbf{x},s^-) Q_{do(\pi,s^-)}(\mathbf{x},d)\right]\right)$, where the expectation is over the state visitation distribution and $\alpha = 2\left(\mathbb{E}[V_{do(\pi,s^+)}(\mathbf{x}_0)] - \mathbb{E}[V_{do(\pi,s^-)}(\mathbf{x}_0)]\right)$. A detailed derivation is included in Appendix A.

**Policy Optimization**. Our choice for policy optimization is mostly typical as a variant of Proximal Policy Optimization (PPO) that incorporates the KL divergence as a penalty Schulman et al. (2017). Using a slight variation of notations, we use $\mathcal{V}_{do(\pi,s)}(\mathbf{x})$ and $\mathcal{Q}_{do(\pi,s)}(\mathbf{x})$ to denote the state/action value function in terms of utility $r_t = r(\mathbf{x}_t, d_t)$. The advantage function for a state is given as $\mathcal{A}_{do(\pi,s)}(\mathbf{x},d) = \mathcal{Q}_{do(\pi,s)}(\mathbf{x},d) - \mathcal{V}_{do(\pi,s)}(\mathbf{x})$. We then write the objective function as

$$J^{PPO}(\theta) = L^{UTIL} - \beta^{KL} L^{KL}$$

where $L^{UTIL} = \hat{\mathbb{E}}_t\left[\frac{\pi(d_t|\mathbf{x}_t,s_t)}{\pi_{old}(d_t|\mathbf{x}_t,s_t)}\hat{\mathcal{A}}_{(\pi,s_t)}(\mathbf{x}_t,d_t)\right]$ is the expected advantage with importance sampling, and $L^{KL} = \hat{\mathbb{E}}_t\left[KL[\pi_{old}(s_t|\mathbf{x}_t,d_t)||\pi(s_t|\mathbf{x}_t,d_t)]\right]$ is the KL divergence to penalize large divergences between the new and old policies. We use the hat symbol to denote the emperical estimation of a given function or operator.

As PPO updates the policy using minibatch SGD, we can separate the batch of timesteps $t$ that were sampled for the minibatches into those belonging to the advantaged group $t^+$ and those for the disadvantaged group $t^-$. The constraint then becomes $\hat{C}_\pi =$

$$\hat{\mathbb{E}}_{t^+}\left[\frac{\pi(d_{t^+}|\mathbf{x}_{t^+},s_{t^+})}{\pi_{old}(d_{t^+}|\mathbf{x}_{t^+},s_{t^+})}\hat{Q}_{do(\pi_{old},s^+)}(\mathbf{x}_{t^+},d_{t^+})\right] - \hat{\mathbb{E}}_{t^-}\left[\frac{\pi(d_{t^-}|\mathbf{x}_{t^-},s_{t^-})}{\pi_{old}(d_{t^-}|\mathbf{x}_{t^-},s_{t^-})}\hat{Q}_{do(\pi_{old},s^-)}(\mathbf{x}_{t^-},d_{t^-})\right].$$

We then integrate the constraint into the full objective function as

$$J(\theta) = L^{UTIL} - \beta^{KL} L^{KL} - \beta^C(\hat{C}_\pi)^2. \tag{2}$$

### 3.4 Causal Decomposition of $C_\pi(\theta)$

As mentioned earlier, it is important to study the connection between long-term and short-term fairness requirements as they may be conflicting objectives. To this end, in this section, we perform a causal analysis to decompose the qualification gain parity, enabling us to identify and distinguish various components contributing to bias. We study the various ways in which the policy $\pi$ influences the qualification gain, where we examine the instantaneous impact of the policy as well as the delayed impact through the transitions of the environment.

As shown in Fig. 1, from a causal perspective, the instantaneous impact of the policy can be captured by the direct edges from $(\mathbf{x}_t, d_t)$ to $g_t$ while the delayed impact can be captured by all other paths from the states to the qualification gain. To examine these different mechanisms through which the policy impact manifests respectively, we leverage the path-specific technique Avin et al. (2005) in causal inference, which can isolate the effect transmitted through a particular causal path while controlling the effect transmitted through other pathways. To achieve this, we construct two hypothetical policies that are not actually implemented. One is a baseline policy $\pi_0$ that always makes the non-treatement decision, i.e., $\pi_0(d^0|\mathbf{x},s) = 1$ (see Fig. 2(a)). The state value function of $\pi_0$, denoted as $V_{do(\pi_0,s)}$, is given by

$$V_{do(\pi_0,s)}(\mathbf{x}) = \sum_{\mathbf{x}'} P(\mathbf{x}'|\mathbf{x},d^0,s)\left(g^s(\mathbf{x},\mathbf{x}') + V_{do(\pi_0,s)}(\mathbf{x}')\right). \tag{3}$$

The other is a virtual policy $\pi^{PS}$ where the qualification gain is assumed to be obtained as if following policy $\pi_0$, while the state transitions occur as if under policy $\pi$ (see Fig. 2(b)). Its state value

function, denoted as $V_{do(\pi^{PS},s)}$, can be given by

$$V_{do(\pi^{PS},s)}(\mathbf{x}) = \sum_{\mathbf{x}'} P(\mathbf{x}'|\mathbf{x},d^0,s)g^s(\mathbf{x},\mathbf{x}') + \sum_d \pi(d|\mathbf{x},s) \sum_{\mathbf{x}'} P(\mathbf{x}'|\mathbf{x},d,s)V_{do(\pi^{PS},s)}(\mathbf{x}'). \quad (4)$$

Facilitated by these hypothetical policies, we can decompose $V_{do(\pi,s)}$ into three components: the component that captures the direct impact of the policy $\pi$ on the qualification gain, the component that captures the delayed impact, as well as the component that captures the spurious effect solely attributable to the environment, given below.

$$V_{do(\pi,s)}(\mathbf{x}) = \underbrace{V_{do(\pi,s)}(\mathbf{x}) - V_{do(\pi^{PS},s)}(\mathbf{x})}_{\text{direct impact}} + \underbrace{V_{do(\pi^{PS},s)}(\mathbf{x}) - V_{do(\pi_0,s)}(\mathbf{x})}_{\text{delayed impact}} + \underbrace{V_{do(\pi_0,s)}(\mathbf{x})}_{\text{spurious effect}}$$

The explanation of this decomposition is as follows. For deriving the direct impact, we employ the policy $\pi^{PS}$ as the reference policy and assume that the decision-making system shifts from this reference policy to policy $\pi$. During this process, the policy $\pi$ is used for generating episodes. However, when calculating the instantaneous qualification gain at each time step, the policy that is used to make the decision shifts from $\pi_0$ to $\pi$, which subsequently influences the next state that is used to compute the qualification gain via $g^s(\mathbf{x}_t, \mathbf{x}_{t+1})$. The direct impact is then computed as the difference in the qualification gain. Since the episodes remain unchanged during the shift, this difference eliminates the policy's delayed impact, capturing only the direct effect of switching from the baseline policy $\pi_0$ to the behavior policy $\pi$.

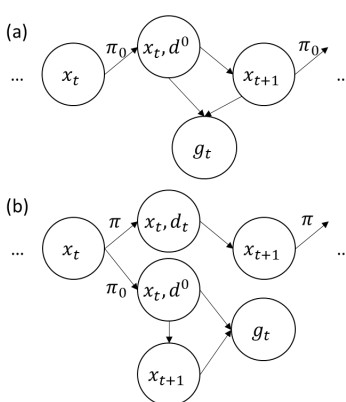

Figure 2: Causal graphs for representing the two hypothetical policies (the reward is omitted from the figures). (a) The baseline policy $\pi_0$ that always makes the negative decisions. (b) The virtual policy where the state transitions occur as if under $\pi$ while the qualification is obtained as if under $\pi_0$.

Similarly, for the delayed impact, we employ the policy $\pi_0$ as the reference policy where both the episodes and the instantaneous qualification gain are determined by $\pi_0$. Then, we switch to policy $\pi^{PS}$, allowing policy $\pi$ to generate the episodes. Thus, the change in the qualification gain is due to the delayed impact of policy $\pi$. Finally, since $\pi_0$ is unrelated to $\pi$, we can treat the qualification gain achieved, i.e., $V_{do(\pi_0,s)}(\mathbf{x})$, as being solely due to the environment's transitions, which can be treated as the spurious effect in the impact of policy $\pi$.

Correspondingly, by taking $S$ into account and conducting the decomposition for both $do(s^+)$ and $do(s^-)$, we can decompose $C_\pi(\theta)$ into three components, referred to as the Direct Policy Effect (DPE), the Indirect Policy Effect (IPE), and the Spurious Policy Effect (SPE), as given below.

$$C_\pi(\theta) = DPE + IPE + SPE,$$

where $DPE = \mathbb{E}\left[V_{do(\pi,s^+)}(\mathbf{x}_0) - V_{do(\pi^{PS},s^+)}(\mathbf{x}_0)\right] - \mathbb{E}\left[V_{do(\pi,s^-)}(\mathbf{x}_0) - V_{do(\pi^{PS},s^+)}(\mathbf{x}_0)\right]$, $IPE = \mathbb{E}\left[V_{do(\pi^{PS},s^+)}(\mathbf{x}_0) - V_{do(\pi_0,s^+)}(\mathbf{x}_0)\right] - \mathbb{E}\left[V_{do(\pi^{PS},s^+)}(\mathbf{x}_0) - V_{do(\pi_0,s^-)}(\mathbf{x}_0)\right]$, and $SPE = \mathbb{E}\left[V_{do(\pi_0,s^+)}(\mathbf{x}_0)\right] - \mathbb{E}\left[V_{do(\pi_0,s^-)}(\mathbf{x}_0)\right]$.

### 3.5 CONNECTION WITH BENEFIT FAIRNESS

The decomposition of $C_\pi(\theta)$ allows us to build a connection between qualification gain parity and instantenous fairness requirments. For the former, we pay special attention to the DPE which represents disparity in the instantaneous impact of the policy on the qualification gain. For the latter, we consider benefit fairness proposed in Plecko & Bareinboim (2023), which does not merely allocate equal treatment to each group's entire population but instead it takes into account the proportion of the population that can benefit from the treatment, making it more suitable for controlling fairness in the outcomes of a decision model.

In Plecko & Bareinboim (2023), the benefit is defined as the conditional average treatment effect (CATE) which represents the increase in the outcome (e.g., survival) associated with the treatment. In our context, the outcome of interest is the qualification gain an individual receives as a result of the treatment. Hence, the benefit can be defined as the expected increase in the qualification gain an individual can achieve if they receive treatment compared to not receiving treatment, i.e.,

$$\Delta(\mathbf{x}, s) = \sum_{\mathbf{x}'} \left( P(\mathbf{x}'|\mathbf{x}, d^1, do(s)) - P(\mathbf{x}'|\mathbf{x}, d^0, do(s)) \right) g^s(\mathbf{x}, \mathbf{x}'),$$

where $P(\mathbf{x}'|\mathbf{x}, d, do(s)) = P(\mathbf{x}'|\mathbf{x}, d, s)$ in our context. Then, benefit fairness is defined as follows.

**Definition 2 (Benefit Fairness Plecko & Bareinboim (2023))** *We say that a policy $\pi$ satisfies benefit fairness if*

$$\forall \mathbf{x}, \mathbf{x}', \pi(d^1|\mathbf{x}, s^+) = \pi(d^1|\mathbf{x}', s^-) \text{ if } \Delta(\mathbf{x}, s^+) = \Delta(\mathbf{x}', s^-).$$

In words, benefit fairness means that individuals from different groups who would benefit equally from a treatment should have similar probabilities of receiving that treatment.

The following proposition reveals the connection between the direct impact component of $V_{do(\pi,s)}(\mathbf{x})$ and benefit. Please refer to Appendix B for the proof.

**Proposition 1** *Given a policy $\pi$ and hypothetical policies $\pi_0, \pi^{PS}$ defined in Eqs. (3), (4), we have*

$$V_{do(\pi,s)}(\mathbf{x}) - V_{do(\pi^{PS},s)}(\mathbf{x}) = \sum_{\mathbf{x}'} \eta^{\pi,s}(\mathbf{x} \to \mathbf{x}')\pi(d^1|\mathbf{x}, s)\Delta(\mathbf{x}, s),$$

$$V_{do(\pi^{PS},s)}(\mathbf{x}) - V_{do(\pi_0,s)}(\mathbf{x}) = \sum_{\mathbf{x}'} (\eta^{\pi,s}(\mathbf{x} \to \mathbf{x}') - \eta^{\pi_0,s}(\mathbf{x} \to \mathbf{x}')) \sum_{\mathbf{x}''} P(\mathbf{x}''|\mathbf{x}, d^0, s)g^s(\mathbf{x}, \mathbf{x}''),$$

*where $\eta^{\pi,s}(\mathbf{x} \to \mathbf{x}')$ is the probability of transitioning from state $\mathbf{x}$ to state $\mathbf{x}'$ with policy $\pi$ after any number of steps.*

Using this proposition, we can reformulate DPE as follows

$$DPE = \mathop{\mathbb{E}}_{\mathbf{x} \sim \pi|s^+} \left[ \pi(d^1|\mathbf{x}, s^+)\Delta(\mathbf{x}, s^+) \right] - \mathop{\mathbb{E}}_{\mathbf{x} \sim \pi|s^-} \left[ \pi(d^1|\mathbf{x}, s^-)\Delta(\mathbf{x}, s^-) \right],$$

where the expectation is over the state visitation distributions.

By combining the above expression and Definition 2, we observe that the DPE is closely related to benefit fairness. When the state visitation distributions in terms of benefit are identical across the two groups, benefit fairness will result in a zero DPE. This implies that both groups have equal opportunities and exposure to the decision-making process so that achieving benefit fairness leads to equitable direct impact. However, if the state visitation distributions differ between the two groups, achieving benefit fairness becomes incompatible with eliminating the DPE. This discrepancy arises because differing state visitation distributions indicate that the two groups are not equally represented in the sequential decision-making process. Hence, enforcing benefit fairness, which aims to equalize the true benefits received by different groups, may inadvertently introduce or maintain disparities in the DPE. On the other hand, we note that the IPE is less sensitive to benefit fairness. Thus, enforcing benefit fairness may not result in a significant change in the IPE. This analysis highlights the complexity of balancing fairness objectives and the challenges of ensuring equitable outcomes across diverse populations in sequential decision-making systems.

**Remark**. A key insight we gain from the above analysis is that, if the benefit $\Delta$ is independent of the sensitive feature $S$, then achieving benefit fairness aligns with decreasing DPE and also loosely corresponds to the goal of equalizing feature distributions across different groups. This is because, in this context, the state visitation distribution approximates the feature distribution, suggesting that benefit fairness could be achieved simultaneously with a reduced DPE and equalized feature distribution. This insight motivates us to introduce an additional consideration into the design qualification gain function when designing practical decision-making systems, which is to ensure that the benefit $\Delta$ remains independent of $S$, even though the transition probability may be dependent on $S$. This may imply that the qualification gain function should be adjusted to account for differences in benefits. It also underscores the importance of understanding why benefits differ between groups, as highlighted in Plecko & Bareinboim (2023).

**Balancing qualification gain parity and benefit fairness**. We propose a simple yet effective approach to achieve benefit fairness while promoting parity in qualification gains during policy optimization. Our approach leverages the concept of individual fairness Dwork et al. (2012) which demands that any two similar individuals should receive similar decision outcomes. Individual fairness is relevant and applicable to our setting, as we require that if two individuals from different groups receive similar benefits from the policy, their probabilities of receiving a positive decision should also be similar. To establish a quantitative metric for benefit fairness, we draw inspiration from the Gini coefficient, commonly used to measure income equality in economic theory Mota et al. (2021), which has also been applied to individual fairness Sirohi et al. (2024). Mathematically, the Geni coefficient is defined as $\sum_{i=1}^{n} \sum_{j=1}^{n} |s_i - s_j| / 2n \sum_{j=1}^{n} s_j$ where $s_i$ is the income of a person. Inspired by this, we define a metric that measures the difference in positive decision rates for all pairs of individuals from different sensitive groups, weighted by the inverse distance in their benefits, as follows.

$$\Lambda = \sum_{\mathbf{x}, \mathbf{x}'} \epsilon \cdot \frac{|\pi(d^1|\mathbf{x}, s^+) - \pi(d^1|\mathbf{x}', s^-)|}{\epsilon + |\Delta(\mathbf{x}, s^+) - \Delta(\mathbf{x}', s^-)|} P(\mathbf{x}|s^+) P(\mathbf{x}'|s^-).$$

Here, $\epsilon$ is a small positive value that prevents division by zero and also controls the radius within which we require similar positive decision rates. As $\epsilon$ decreases, similar positive decision rates are only enforced when the benefits are nearly identical. Conversely, a large $\epsilon$ will penalize differences in positive decision rates when the benefits are close but not exactly the same. The final objective function is obtained by incorporating $\Lambda$ into Eq. (2):

$$J(\theta) = L^{UTIL} - \beta^{KL} L^{KL} - \beta^C (\hat{C}_\pi)^2 - \beta^\Lambda \Lambda. \tag{5}$$

## 4 EXPERIMENTS

We conduct experiments to evaluate the policy optimization algorithms we have proposed and compare them with baselines regarding the achievement of long-term fairness[1]. We also present the results from the decomposition and the performance of our algorithm when taking benefit fairness into consideration, demonstrating how the constraint of benefit fairness influences different components of qualification parity. We refer to our algorithm with objective function Eq. (2) as PPO-C and the algorithm objective function Eq. (5) as PPO-Cb.

### 4.1 EXPERIMENTAL SETUP

We leverage the simulation environment developed in D'Amour et al. (2020) that is commonly used in related work (e.g., Yu et al. (2022); Hu et al. (2023)). The environment is designed to simulate the process of a bank dispersing loans to members of a population. Each individual in the population belongs to either the advantaged group $s^+$ or the disadvantaged group $s^-$. At each time step, an individual applies for a loan and the bank has to make a binary decision, $d_t \in \{d^0, d^1\}$, on whether to approve or deny the loan. The decision is made in accordance with the current policy $\pi_\theta(d|s_t, \mathbf{x}_t)$, a distribution produced by a feedforward neural network from which the decision $d_t$ is sampled. By default, the average qualification level of individuals belonging to $s+$ is higher than that of those belonging to $s-$. These distributions evolve with time in response to factors in the environment and loan decisions.

To facilitate the idea that the environment can have effects, unrelated to the decision, on an individual's qualification level, we also include a term $x_{drift} \in \{-1, 0, 1\}$. This represents the change in an individual's credit score which would occur regardless of whether or not a loan was received by the individual. As a result, an individual is represented by the 4-tuple $(s, \mathbf{x}, x_{drift}, y)$. To sample an individual, first we sample the sensitive attribute $s \sim P(s)$, then the credit score $\mathbf{x} \sim P(\mathbf{x}_t|s)$, followed by the ability to repay the loan $y \sim P(y|\mathbf{x}, s)$. The credit drift $x_{drift} \sim P(x_{drift})$ is sampled independently of the other attributes. Note that $x_{drift}$ and $y$ are unobserved for the policy and are not used as input for decision-making.

We designed the qualification gain function to reflect the real-world phenomenon where progressing from beginner to intermediate requires less effort than progressing from advanced to expert. The

---

[1] All the code is available at `https://github.com/j-proj/Causal-Lens-Fair-RL`.

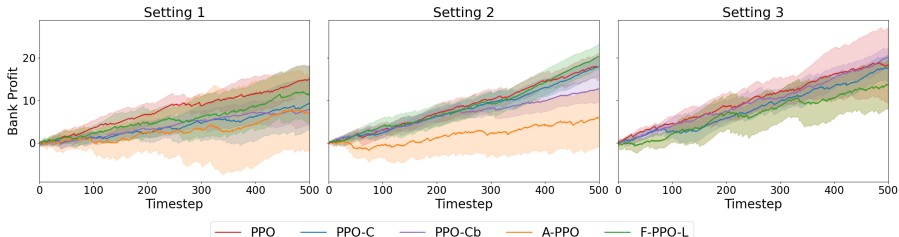

Figure 3: Utility comparison across different settings.

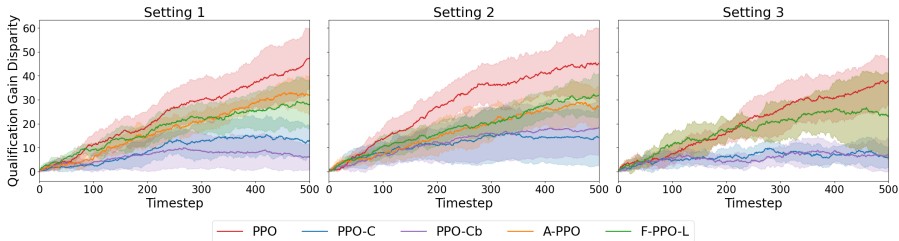

Figure 4: Qualification gain disparity comparison across different settings.

distribution $P(\mathbf{x}_{t+1}|\mathbf{x}_t, s)$ is a deterministic function of $(s_t, \mathbf{x}_t, x_{drift}, y_t, d_t)$ where a small amount of mass representing an individual is moved from $P(\mathbf{x}_t|s_t)$ to $P(\mathbf{x}_{t+1}|s_t)$. In the evaluation, we use three different settings for the simulation environment which we will simply refer to as Setting 1,2, and 3. These settings differ from each other in initial distributions over credit scores, repayment probabilities, and credit drift likelihoods. The repayment probabilities are learned by using the Home Credit Default Risk dataset Montoya et al. (2018) and a dataset previously released by Lending Club Wagh (2017). The details of the experimental setup are provided in Appendix C.

We compare our methods to three baselines: (1) PPO Schulman et al. (2017), a variant of the vanilla PPO algorithm incorporating a KL-divergence penalty; (2) A-PPO Yu et al. (2022), which introduces additional regularizations to penalize the advantage function in order to enforce equalized opportunity; and (3) F-PPO-L Hu et al. (2023), which also applies a regularization term to penalize the advantage function, but with the objective of reducing the 1-Wasserstein distance between group feature distributions.

Due to the large variance in a model's performance in both training and evaluation due to the highly stochastic nature of the system, for all results and methods, we perform multiple training runs with different random seeds for initialization and then take an average to better gauge performance.

## 4.2 RESULTS

**Evaluating long-term fairness**. We first evaluate the effectiveness of our proposed approaches in achieving qualification gain parity. We compare PPC-C and PPC-Cb against the baseline methods PPO, which purely optimizes for utility, as well as A-PPO and F-PPO-L which optimize for utility and their own fairness objectives. In Figure 3, we see that PPO-C, the constraint-only variant, is fairly competitive with regard to utility when compared to the baselines. While the inclusion of the additional benefit fairness term in PPO-Cb tends to decrease the performance, it manages to produce a respectable profit in Setting 2, where A-PPO struggles to do so. Since PPO-C and PPO-Cb both optimize for the qualification gain disparity, it is unsurprising that they both outperform the baselines in this regard as shown in Figure 4. However, it is interesting that PPO-Cb outperforms PPO-C in Setting 1, while in Settings 2 and 3 they exhibit relatively similar performance. This may be attributed to the connection between qualification gain disparity and benefit fairness, which is further analyzed in the following.

**Evaluating causal decomposition**. We then evaluate the causal decomposition and how different components are impacted in the training. In Figure 5, we show $C_\pi(\theta)$ together with the decomposition of the constraint into its constituent components that are IPE, DPE, and SPE for PPO-C and

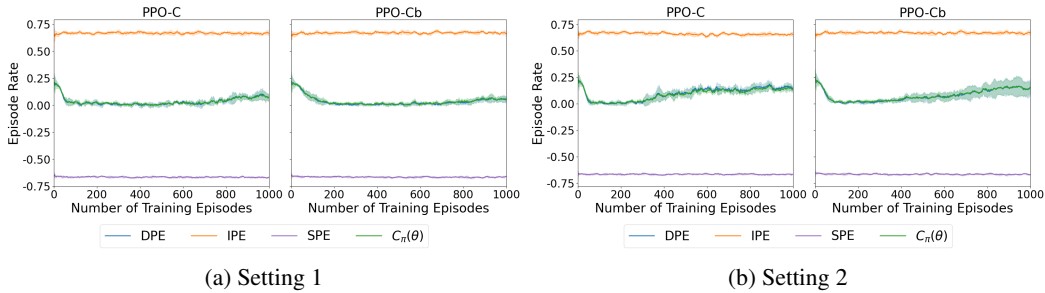

Figure 5: Qualification gain disparity decompositions.

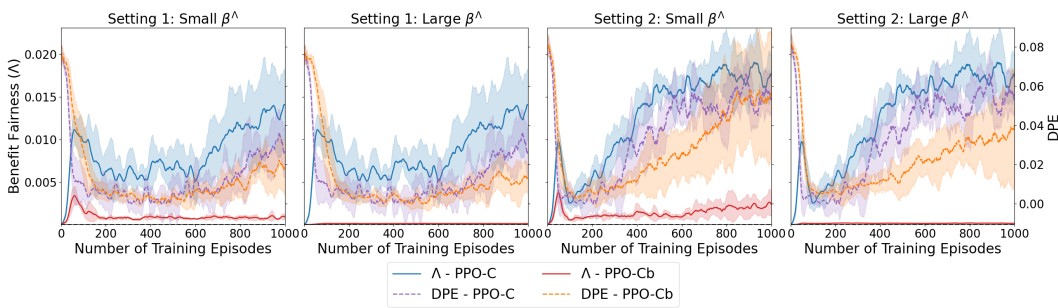

Figure 6: Benefit fairness and DPE for PPO-C and PPO-Cb model variants.

PPO-Cb. In both settings, the DPE varies more significantly over time compared with the IPE. This behavior implies that policy optimization is more effective in decreasing DPE, the instantaneous impact of the policy, than IPE, the delayed impact of the policy. Comparing the two methods, PPO-Cb exhibits a more pronounced reduction in the DPE than in the IPE, suggesting an inherent connection between the DPE and benefit fairness.

**Evaluating benefit fairness**. Finally, we evaluate the effectiveness of objective function Eq. (5) in ensuring benefit fairness as well as the connection between benefit fairness and DPE. Figure 6 is a demonstration of the effect of increasing the $\beta^\Lambda$ parameter to increase the enforcement of the benefit fairness constraint in PPO-Cb, with PPO-C included as further reference. The first two plots are generated using the environment Setting 1, with the left corresponding to the base setting for $\beta^\Lambda$ and the right for the relatively large setting. Comparing the two plots, we see that benefit fairness improves as $\beta^\Lambda$ increases, showing the effecacy of the regularization. Moreover, we see that the DPE decreases for a larger $\beta^\Lambda$, which further reflects the connection between benefit fairness and DPE as demonstrated in our theoretical analysis. This pattern similarly manifests in the last two figures, where Setting 2 is used.

## 5 CONCLUSIONS

In this paper, we explored the achievement of long-term fairness in reinforcement learning (RL) from a causal perspective, emphasizing the importance of simultaneously addressing instantaneous fairness requirements. We proposed a general RL framework where long-term fairness is quantified by the difference in the average expected qualification gain that individuals from different groups could obtain. Through a causal decomposition of this disparity, we identified three key components: the Direct Policy Effect (DPE), the Indirect Policy Effect (IPE), and the Spurious Policy Effect (SPE). Our analysis revealed an intrinsic connection between benefit fairness—an emerging short-term fairness concept—and DPE, which is particularly sensitive to policy optimization. Furthermore, we developed a simple yet effective approach to balance qualification gain parity with benefit fairness. Experimental results demonstrated the efficacy of our methods. In future work, we aim to conduct a more in-depth analysis of the qualification gain function's design to account for differences in benefits and align various fairness notions in sequential decision-making.

ACKNOWLEDGMENTS

This work was supported in part by NSF 1910284, 2142725.

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

APPENDIX A: DERIVING $\nabla_\theta C_\pi(\theta)$

To derive $\nabla_\theta V_{do(\pi,s)}(\mathbf{x})$, we have

$$\nabla_\theta V_{do(\pi,s)}(\mathbf{x}) = \nabla_\theta \sum_d \pi(d|s,\mathbf{x}) Q_{do(\pi,s)}(\mathbf{x},d)$$

$$= \sum_d \left( \nabla_\theta \pi(d|s,\mathbf{x}) Q_{do(\pi,s)}(\mathbf{x},d) + \pi(d|s,\mathbf{x}) \nabla_\theta Q_{do(\pi,s)}(\mathbf{x},d) \right)$$

$$= \sum_d \left( \nabla_\theta \pi(d|s,\mathbf{x}) Q_{do(\pi,s)}(\mathbf{x},d) + \pi(d|s,\mathbf{x}) \nabla_\theta \sum_{\mathbf{x}'} P(\mathbf{x}'|\mathbf{x},d,s) \left( g(\mathbf{x},\mathbf{x}') + V^\pi(s,\mathbf{x}') \right) \right)$$

$$= \sum_d \left( \nabla_\theta \pi(d|s,\mathbf{x}) Q_{do(\pi,s)}(\mathbf{0},d) + \pi(d|s,\mathbf{x}) \sum_{\mathbf{x}'} P(\mathbf{x}'|\mathbf{x},d,s) \nabla_\theta V^\pi(s,\mathbf{x}') \right)$$

If we let $\phi(\mathbf{x}) = \sum_d \nabla_\theta \pi(d|s^+,\mathbf{x}) Q_{do(\pi,s)}(\mathbf{x},d)$, then

$$\nabla_\theta V_{do(\pi,s)}(\mathbf{x}) = \phi(\mathbf{x}) + \sum_d \pi(d|s,\mathbf{x}) \sum_{\mathbf{x}'} P(\mathbf{x}'|\mathbf{x},d,s) \nabla_\theta V^\pi(s,\mathbf{x}')$$

$$= \phi(\mathbf{x}) + \sum_{\mathbf{x}'} \sum_d \pi(d|s,\mathbf{x}) P(\mathbf{x}'|\mathbf{x},d,s) \nabla_\theta V^\pi(s,\mathbf{x}')$$

$$= \phi(\mathbf{x}) + \sum_{\mathbf{x}'} \rho^\pi(\mathbf{x} \to \mathbf{x}', 1) \nabla_\theta V^\pi(s,\mathbf{x}')$$

$$= \phi(\mathbf{x}) + \sum_{\mathbf{x}'} \rho^{\pi,s}(\mathbf{x} \to \mathbf{x}', 1) \left( \phi(\mathbf{x}') + \sum_{\mathbf{x}''} \rho^{\pi,s}(\mathbf{x}' \to \mathbf{x}'', 1) \nabla_\theta V^\pi(s,\mathbf{x}'') \right)$$

$$= \phi(\mathbf{x}) + \sum_{\mathbf{x}'} \rho^{\pi,s}(\mathbf{x} \to \mathbf{x}', 1) \phi(\mathbf{x}') + \sum_{\mathbf{x}''} \rho^{\pi,s}(\mathbf{x} \to \mathbf{x}'', 2) \nabla_\theta V^\pi(s,\mathbf{x}'')$$

$$= \cdots$$

$$= \sum_{\mathbf{x}^*} \sum_{k=0}^\infty \rho^{\pi,s}(\mathbf{x} \to \mathbf{x}^*, k) \phi(\mathbf{x}^*)$$

By denoting $\eta^{\pi,s}(\mathbf{x} \to \mathbf{x}^*) = \sum_{k=0}^\infty \rho^{\pi,s}(\mathbf{x} \to \mathbf{x}^*, k)$, then we can rewrite $\nabla_\theta V^\pi(s,\mathbf{x})$ as $\sum_{\mathbf{x}^*} \eta^{\pi,s}(\mathbf{x} \to \mathbf{x}^*) \phi(\mathbf{x}^*)$.

Plugging these results into $\nabla_\theta C_\pi(\theta)$ for $s^+, s^-$, we have that

$$\nabla_\theta C_\pi(\theta) = \alpha \left( \mathbb{E}_{\pi|s^+} \left[ \nabla_\theta \ln \pi(d|\mathbf{x},s^+) Q_{do(\pi,s^+)}(\mathbf{x},d) \right] - \mathbb{E}_{\pi|s^-} \left[ \nabla_\theta \ln \pi(d|\mathbf{x},s^-) Q_{do(\pi,s^-)}(\mathbf{x},d) \right] \right)$$

where

$$\alpha = 2 \left( \mathbb{E}[V_{do(\pi,s^+)}(\mathbf{x}_0)] - \mathbb{E}[V_{do(\pi,s^-)}(\mathbf{x}_0)] \right).$$

APPENDIX B: PROOF OF PROPOSITION 1

Based on the definition, we have that

$$V_{do(\pi,s)}(\mathbf{x}) = \sum_d \pi(d|\mathbf{x},s) \sum_{\mathbf{x}'} P(\mathbf{x}'|\mathbf{x},d,s) \left( g^s(\mathbf{x},\mathbf{x}') + V_{do(\pi,s)}(\mathbf{x}') \right),$$

and

$$V_{do(\pi^{PS},s)}(\mathbf{x}) = \sum_{\mathbf{x}'} P(\mathbf{x}'|\mathbf{x},d^0,s) g^s(\mathbf{x},\mathbf{x}') + \sum_d \pi(d|\mathbf{x},s) \sum_{\mathbf{x}'} P(\mathbf{x}'|\mathbf{x},d,s) V_{do(\pi^{PS},s)}(\mathbf{x}').$$

It follows that

$$V_{do(\pi,s)}(\mathbf{x}) - V_{do(\pi^{PS},s)}(\mathbf{x})$$

$$= G^{\pi,s}(\mathbf{x}) + \sum_d \pi(d|\mathbf{x},s) \sum_{\mathbf{x}'} P(\mathbf{x}'|\mathbf{x},d,s) \left( V_{do(\pi,s)}(\mathbf{x}') - V^{PS}_{do(\pi,s)}(\mathbf{x}') \right)$$

where

$$G^{\pi,s}(\mathbf{x}) = \sum_{\mathbf{x}'} \left( \sum_d \pi(d|\mathbf{x},s)P(\mathbf{x}'|\mathbf{x},d,s)g^s(\mathbf{x},\mathbf{x}') - P(\mathbf{x}'|\mathbf{x},d^0,s)g^s(\mathbf{x},\mathbf{x}') \right)$$

By writing $P(\mathbf{x}'|\mathbf{x},d^0,s)g^s(\mathbf{x},\mathbf{x}')$ as $(\pi(d^1|\mathbf{x},s)+\pi(d^0|\mathbf{x},s)P(\mathbf{x}'|\mathbf{x},d^0,s)g^s(\mathbf{x},\mathbf{x}'))$ and rearrange the above expression, we have

$$G^{\pi,s}(\mathbf{x}) = \pi(d^1|\mathbf{x},s)\sum_{\mathbf{x}'}\left(P(\mathbf{x}'|\mathbf{x},d^1,s) - P(\mathbf{x}'|\mathbf{x},d^0,s)\right)g^s(\mathbf{x},\mathbf{x}') = \pi(d^1|\mathbf{x},s)\Delta(\mathbf{x},s)$$

By similarly following the policy gradient theorem, we have

$$V_{do(\pi,s)}(\mathbf{x}) - V_{do(\pi^{PS},s)}(\mathbf{x}) = \sum_{\mathbf{x}'} \eta^{\pi,s}(\mathbf{x}\to\mathbf{x}')G_{\pi,s}(\mathbf{x}')$$

The second equation in the proposition can be similarly derived.

## APPENDIX C: EXPERIMENTAL SETUP DETAILS

**Qualification gain function**. We would like the qualification gain function to reflect real-world phenomena to some extent. One viewpoint is that as an individual's qualification level increases, further progression becomes increasingly more difficult. Progression from beginner to intermediate takes less effort than progression from advanced to expert, so the qualification gain at high levels should have more weight. To encapsulate this idea, the qualification gain function for a single transition is defined as

$$g(\mathbf{x}_t, \mathbf{x}_{t+1}) = \frac{\mathbf{x}_{t+1}^3 - \mathbf{x}_t^3}{\left| \max_{|i-j|=1} g(x_i, x_j) \right|}$$

where the denominator serves to reduce the range of possible values.

**Transition probability**. The distribution $P(\mathbf{x}_{t+1}|\mathbf{x}_t, s)$ is a deterministic function of $(s_t, \mathbf{x}_t, x_{drift}, y_t, d_t)$ where a small amount of mass representing an individual is moved from $P(\mathbf{x}_t|s_t)$ to $P(\mathbf{x}_{t+1}|s_t)$, i.e.,

$$\mathbf{x}_{t+1} = \begin{cases} \mathbf{x}_t + x_{drift} + 1 & \text{if}(d_t = 1, y_t = 1), \\ \mathbf{x}_t + x_{drift} - 1 & \text{if}(d_t = 1, y_t = 0), \\ \mathbf{x}_t + x_{drift} & \text{if}(d_t = 0). \end{cases}$$

**Enviroment settings**. In the evaluation of our proposed algorithms, we use three different settings for the simulation environment which we will simply refer to as Setting 1,2, and 3. These settings differ from each other in initial distributions over credit scores, repayment probabilities, and credit drift likelihoods.

We generated the repayment probabilities by fitting a logistic regression model to credit score datasets and used the predicted repayment probability from each credit score level. Setting 1 and 2 both use the same distribution over initial credit scores where $\mathbb{E}_{\mathbf{x}_0\sim s^-}[\mathbf{x}_0] < \mathbb{E}_{\mathbf{x}_0\sim s^+}[\mathbf{x}_0]$. They differ in the repayment probabilities where Setting 1 uses probabilities generated using the Home Credit Default Risk dataset Montoya et al. (2018), and the probabilities for Setting 2 are from a dataset previously released by Lending Club Wagh (2017), a type of peer-to-peer lending market.

In Setting 3, however, the initial distribution for both groups is the same. What does differ between groups are the credit drift probabilities $p(x_{drift})$, which is now dependent on the sensitive attribute. Here, $p(x_{drift} = -1|s^-) > p(x_{drift} = -1|s^+)$ while $p(x_{drift} = 1|s^-) < p(x_{drift} = 1|s^+)$.

## APPENDIX D: INFLUENCES OF $\beta^{KL}$ AND $\beta^\Lambda$

We further examine how the penalty coefficients $\beta^{KL}$ and $\beta^\Lambda$ quantitatively impact fairness and model performance. As shown in Figure 7, a larger $\beta^{KL}$ can help reduce the model variance. In Figure 8, we show the influence of the strength of enforcing benefit fairness on long-term fairness. Notably, we observe that the loan rate for $\beta^\Lambda = 2$ is more balanced than that for $\beta^\Lambda = 0$, further implying that benefit fairness may align with long-term fairness in certain conditions.

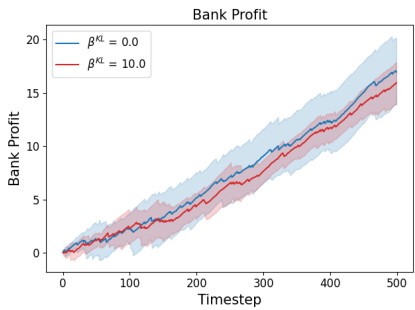

Figure 7: Utility earned by the PPO-C variant comparing when the agent was trained with $\beta^{KL} = 0$ versus $\beta^{KL} = 10$.

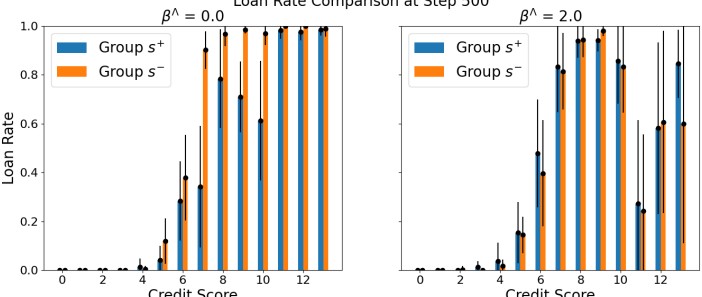

Figure 8: Lending rates for the PPO-Cb variant in setting 2.

