# OpenReview forum: "A Causal Lens for Learning Long-term Fair Policies"
_ICLR.cc/2025/Conference — ICLR 2025 Poster_

### Official Review · Reviewer_FkQm · 2024-10-17

**Soundness:** 3
**Presentation:** 3
**Contribution:** 2
**Rating:** 6
**Confidence:** 3

**Summary:**

The paper proposed a general causal framework to characterize long-term fairness in reinforcement learning (RL) settings, where the unfairness is represented by "qualification gain disparity" between the advantaged group and the disadvantaged group ($C_{\pi}$). The authors decomposed the unfairness into three different parts (DPE, IPE, SPE) and used Proximal Policy Optimization (PPO) to optimize the decision policy. Particularly, DPE is closely related to benefit fairness (i.e., individuals from different groups who gain similar qualification improvement should have similar opportunities to get the treatment), and if the benefits are independent of the group attribute, then benefit fairness means with 0 DPE, demonstrating some kind of alignment between the long-term fairness proposed by the authors and benefit fairness. Motivated by this, the authors also added fairness as a separate term to the fair objective function. Thus, there are 2 fairness mechanisms ($PPO-C$, $PPO-Cb$) proposed, and the authors did simulation studies on real datasets to verify their findings.

**Strengths:**

1. Studying Long-term fairness as qualification gain disparity is well-motivated and of social importance.
2. I like the idea of the causal decomposition using $\pi_{PS}$.
3. The experiments demonstrate the effectiveness of the fairness mechanisms.

**Weaknesses:**

Overall, I think the paper is interesting, but so far it does not situate well in previous literature. I am willing to increase my score if the authors clarify their contribution.

1. Though I appreciate the efforts the authors made to compare their work to previous literature, I still feel the novelty of the paper is not clear enough. Firstly, just as the authors stated, [Hu et al., 2023, Hu & Zhang, 2022, Yu et al., 2022] considered long-term fairness in a causal framework. Especially for the framework proposed by [Hu & Zhang, 2022], the main difference seemed to be: (i) they considered the disparity of qualification rate (not the gain of qualification) as the fairness notion; (ii) they did not consider RL settings. Could the authors try making the comparison clearer?

2. There are other recent papers on strategic classification and performative prediction considering long-term fairness in terms of qualification rate improvement (also known as social welfare) when population distribution can be shaped by the decision policy (e.g., [Jia et al., 2024, Jin et al., 2024]). Moreover, there are at least two fairness notions related to qualification gain: (i) equal improvement [Guldogan et al., 2022]; (ii) bounded effort [Heidari et al., 2019]. It is necessary to review these papers.

3. The fairness mechanisms just stack all terms together and I am not sure about the novelty. Plus, in Figure 5, it seems that IPE and SPE are rarely influenced by the proposed fairness mechanisms. Could you explain why?

4. Why not provide error bars for Figure 4?

> References

Jia, Zhuangzhuang, et al. "Distributionally Robust Performative Optimization." arXiv preprint arXiv:2407.01344 (2024).

Jin, Kun, et al. "Addressing Polarization and Unfairness in Performative Prediction." arXiv preprint arXiv:2406.16756 (2024).

Guldogan, Ozgur, et al. "Equal improvability: A new fairness notion considering the long-term impact." arXiv preprint arXiv:2210.06732 (2022).

Heidari, Hoda, Vedant Nanda, and Krishna P. Gummadi. "On the long-term impact of algorithmic decision policies: Effort unfairness and feature segregation through social learning." arXiv preprint arXiv:1903.01209 (2019).

**Questions:**

1. Could you make your contribution clearer and review previous work more comprehensively? (weakness 1, 2)

2. Why are IPE and SPE not influenced by the fairness mechanisms? (weakness 3)

3. Could you justify why it helps by directly adding benefit fairness as another regularization term? (weakness 3)

4. Could you provide error bars for Figure 4, or am I missing something? (weakness 4)

---

> ### Author Response · Authors · 2024-11-21
> **Q1**
>
> Thanks for your insightful comments and for raising this important question. In addition to the RL setting, we think our work shares a nuanced connection with [Hu & Zhang, 2022]. While we address a similar problem to that of [Hu & Zhang, 2022], we employ different methodologies that lead to novel insights. To provide a clearer comparison, let’s first explain the connection between our work and [Hu & Zhang, 2022] and then highlight the differences.
>
> In [Hu & Zhang, 2022], long-term fairness is defined as the difference in qualification rates at a future time point $t^*$, i.e.,
>
>  $\mathbb{E} [ P(y^{t^*}=1|x^{t^*},do(s^+) ] - \mathbb{E} [P(y^{t^*}=1|x^{t^*},do(s^-) ]$.
>
> This definition can be represented using our qualification gain function if we define:
>
> $g^{s}(x^{t},x^{t+1}) = P(y^{t^*}| x^{t+1}, do(s)) - P(y^{t^*}| x^{t}, do(s))$
>
> and introduce a virtual initial time step with a fixed qualification gain $P(y^{t^*}| x^{0}, do(s))$. With this setup, the state value function $V_{do(\pi,s)(x)}$ becomes equal to $\mathbb{E} [ P(y^{t^*}=1|x^{t^*},do(s) ]$. This analysis demonstrates the connection between our proposed qualification gain and the qualification rate used in [Hu & Zhang, 2022].
>
> However, our work employs a different method to analyze the relationship between long-term and short-term fairness. In [Hu & Zhang, 2022], for short-term fairness, they focus on sensitive feature unawareness in the decision (e.g., demographic parity) by analyzing the paths from the sensitive feature to the decision in the causal graph. Their conclusion is that there is an inherent conflict between long-term and short-term fairness.
>
> In contrast, our work reveals a theoretical connection between long-term fairness and benefit fairness by proposing a novel causal decomposition technique. Specifically, we examine the impact of the policy on the $outcome$ of the decision (i.e., the qualification gain in our context), whereas the decision outcome is not explicitly modeled in [Hu & Zhang, 2022] (note that the decision outcome is different from the decision itself).
>
> Regarding our findings, our analysis shows that benefit fairness may not necessarily conflict with long-term fairness, implying the possibility of reconciling short-term and long-term fairness objectives. This contrasts with the conclusions of [Hu & Zhang, 2022], thereby providing new insights into how fairness can be achieved over time without inherent conflicts.
>
> In summary, although we tackle a similar problem as [Hu & Zhang, 2022], our distinct methodologies, including the incorporation of RL settings and the causal decomposition, lead to different conclusions and novel insights. We believe these differences not only highlight the novelty of our work but also contribute valuable perspectives to the ongoing discussion on long-term fairness.
>
> For other references, performative prediction is a different framework focused on model-dependent distribution shifts in machine learning models. Although it is a powerful approach that can offer theoretical convergence guarantees, it has its limitations. For instance, due to its complexity, it often considers decoupled performative risk, where the data distribution is influenced by parameters different from those being optimized. Equal improvability investigates a related but different problem setting, where the model learns to apply additional efforts to improve individuals' qualifications.
>
> After reviewing these papers, we believe that our work offers unique contributions to the field of long-term fair machine learning.

---

> > ### Comment · Reviewer_FkQm · 2024-11-22
> >
> > Thanks for your response. I appreciate your clarification and understand that the novelty mainly lies in the RL setting and the incorporation of benefit fairness. I also thank the authors for updating the error bars. My main concern is that I am not sure the theoretical contribution and novelty are enough, especially when benefit fairness is also a notion in previous work.
> >
> > Moreover, I do not see an updated draft (at least not highlighted in different colors) where the main novelty compared to the previous works is discussed and emphasized. As other reviewers also pointed out, it is necessary to include related works in performative prediction/strategic classification. I understand that works in PP focus more on convergence and model-dependent distribution shifts, but the frameworks are still highly related to this paper. I strongly suggest the authors modify the introduction/related work to highlight their contributions (and also highlight the modification).
> >
> > Currently I will keep my score.

---

> > > ### Author Response · Authors · 2024-11-22
> > >
> > > Thank you for your timely response. We will modify the Introduction and Related Work sections and prepare an updated draft within the rebuttal period.

---

> > > > ### Comment · Reviewer_FkQm · 2024-11-24
> > > >
> > > > Thanks for modifying the draft. My main concerns are resolved. After careful consideration, I think the causal decomposition has some merit and long-term fairness is an important topic in RL. I increase my score to 6.

---

> ### Author Response · Authors · 2024-11-21
> **Q2**
>
> The Spurious Policy Effect (SPE) is inherently independent of the policy $\pi$ and, consequently, remains influenced by fairness mechanisms. Regarding the Indirect Policy Effect (IPE), its lower sensitivity to fairness mechanisms may arise from the inherent uncertainties within the environment, making IPE more challenging to optimize effectively.

---

> ### Author Response · Authors · 2024-11-21
> **Q3**
>
> Regularization is a well-established and easy-to-implement technique for formulating objective functions. It is widely used in the literature, including in the works you cited (e.g., [[Jia et al., 2024, Guldogan et al., 2022]). Regarding our work, one way to interpret it is as a re-weighting of the total reward based on the penalty for violating benefit fairness. This formulation encourages the model to satisfy fairness constraints, leading to more equitable outcomes while still optimizing performance.

---

> ### Author Response · Authors · 2024-11-21
> **Q4**
>
> The reason we didn’t include the error bars in Figures 3 and 4 is that they would make the figures unreadable. We have re-plotted the figures with the error bars, which are shown in Figure 9 in the appendix of the revised PDF of the paper.

---

### Official Review · Reviewer_KhgC · 2024-11-03

**Soundness:** 3
**Presentation:** 2
**Contribution:** 3
**Rating:** 5
**Confidence:** 3

**Summary:**

This paper uses the causal perspective to study the long-term fairness problem where decisions can

**Strengths:**

1. The long-term fairness under causal inference perspective seems novel
2. The fairness penalty decomposition is useful in practice

**Weaknesses:**

1. Theoretical guarantee not sufficient:
(1). Convergence of the algorithm
(2). How the penalty coefficients \beta^{KL} and \beta^{\Lambda} quantitatively impact fairness and model performance (e.g., precision/recall per round)

2. Despite fairness metrics improvement, it is difficult to tell if the model performance improved and if the fairness-aware optimization can result in a Pareto superior result for every group or every individual, there should be more theoretical and numerical evidence on the utilities instead of just utility gaps

3. References:
(1). The long-term fairness problems are studied in a previous line of work related to repeated distributionally robust optimizations, e.g., "Fairness Without Demographics in Repeated Loss Minimization"
(2). The performative predictions and the strategic classification/regression problems are other research areas closely related to model decision having causal effect on data distribution, please compare with them

4. Similarities and differences between the new metrics and the conventional fairness metrics:
It will be more helpful to build connections between the fairness terms in the long-term causal formulation with metrics like demographic parity, equal opportunity, equalized odds in the conventional settings like single shot classification and regression problems, which I think equalized odds (both TPR and FPR being the same) as well as loss disparities do not have counterparts in the causal formulation, and I encourage authors to provide explanations on why not.

**Questions:**

1. Why fairness penalty instead of re-weighting in the optimization problem?

**Details Of Ethics Concerns:**

No concerns on this

---

> ### Author Response · Authors · 2024-11-21
> **W1**
>
> For the penalty coefficients $\beta^{KL}$ and $\beta^{\Lambda}$, the former influences the variance of the model, while the latter affects the strength of enforcing benefit fairness. We have conducted additional experiments to empirically evaluate their effects, as shown in Figures 7 and 8 in the appendix of the revised PDF of the paper. Notably, in Figure 8, we observe that the loan rate for $\beta^{\Lambda} = 2$ is more balanced than that for $\beta^{\Lambda} = 0$, further implying that benefit fairness may not conflict with long-term fairness.

---

> ### Author Response · Authors · 2024-11-21
> **W2**
>
> We agree that our proposed method does not include a formal guarantee for achieving long-term fairness like the Pareto optimality, largely due to the complexity of analyzing the convergence properties of complex reinforcement learning algorithms like PPO.

---

> ### Author Response · Authors · 2024-11-21
> **W3**
>
> Thank you for your comments. We agree that performative prediction is an important framework for studying dynamic systems. However, we would like to clarify that performative prediction is a different framework focused on model-dependent distribution shifts in machine learning models. Although it is a powerful approach that can offer theoretical convergence guarantees for algorithms like Repeated Loss Minimization, it has its limitations. For instance, due to its complexity, it often considers decoupled performative risk, where the data distribution is influenced by parameters different from those being optimized. Compared with performative prediction, we believe that our work also has its own merits in advancing the discussion on long-term fairness.

---

> ### Author Response · Authors · 2024-11-21
> **W4**
>
> The connection between long-term fairness and conventional fairness metrics like demographic parity (DP) and equal opportunity (EO) has been examined in previous theoretical studies, e.g., [1]. These works generally conclude that enforcing DP or EO can negatively impact long-term fairness in typical scenarios. Our analysis, on the other hand, reveals that benefit fairness does not necessarily conflict with long-term fairness, suggesting that it may be feasible to align benefit fairness with long-term objectives.
>
> [1] Zhang, Xueru, et al. "How do fair decisions fare in long-term qualification?." Advances in Neural Information Processing Systems 33 (2020): 18457-18469.

---

> ### Author Response · Authors · 2024-11-21
> **Q1**
>
> We appreciate the reviewer’s question regarding our choice to use a fairness penalty instead of re-weighting in the optimization problem. In fact, incorporating a fairness penalty into the objective function can be viewed as a form of re-weighting applied to the overall reward. Traditional re-weighting methods typically adjust the weights of individual rewards, which can be challenging to implement in our case when distributing fairness constraints across each individual component. Therefore, we opt to use a fairness penalty to effectively integrate fairness constraints directly into the optimization process, ensuring a more cohesive and manageable approach to achieving fairness.

---

### Official Review · Reviewer_hawr · 2024-11-03

**Soundness:** 3
**Presentation:** 3
**Contribution:** 3
**Rating:** 6
**Confidence:** 3

**Summary:**

The paper studies long-term fairness in causal reinforcement learning. The paper proposes the difference between the average qualification gain achieved by each group as their notion of long-term fairness and uses a PPO-type optimization objective to enforce fairness. Through a casual decomposition, the authors break down the qualification gain difference into 3 terms: direct impact, delayed impact, and spurious effects. Empirically, the paper studies how to lower the disparity in average qualification gains across groups and study the effect of each of the three terms.

**Strengths:**

-- The paper studies long-term fairness which is an important, yet understudied problem.

-- The paper proposes a novel fairness metric for long-term fairness in causal reinforcement learning.

**Weaknesses:**

-- There are no technical algorithmic novelties in the paper as the optimization problem is inspired by PPO. This is not a major weakness as the paper introduces novel elements like the qualification gain function and proposes a causal decomposition of the qualification gain.

-- The experimental setup does not clearly show the advantage of the approach over prior baselines. For example, the difference between the worst and best approach is a mere 0.2% in all experiments in Figure 2. Are there other settings where this difference is more significant? The qualification gain disparity in Figure 3, does not go to 0 for each of the proposed metrics and seems to increase with increasing horizon. Can the authors provide why this is the case? Are there other settings where the qualification gain disparity goes to 0?

**Questions:**

-- Can the authors specify how I should interpret the results of Figures 2 and 3 in light of the weaknesses mentioned above?

-- What is the relationship of this notion of fairness, with the ones aiming to equalize rewards across all groups? See https://proceedings.mlr.press/v130/wen21a.html and references within.

---

> ### Author Response · Authors · 2024-11-21
> **Q1**
>
> Thanks for your comments. I believe you are referring to Figures 3 and 4. Figure 3 presents the utility comparison. The absolute value of the utility depends on the utility function. Specifically, in our context, the bank’s initial cash and the interest rate. In our experiments, we set the initial cash to \\$10,000 and the interest rate to 10\%. That is why the difference between the worst and best approach is a mere 0.2%. We can easily increase the relative difference between different methods to more than 100\% by setting the initial cash to \\$0.

---

> ### Author Response · Authors · 2024-11-21
> **Q2**
>
> [1] is one of the earliest works in studying long-term fairness in MDP. While both our work and [1] address fairness in dynamic decision-making systems, there are significant and nuanced differences in the fairness notions we employ. In [1], fairness is defined as the equality of the cumulative "rewards" received by different groups. In contrast, our work decouples the reward from the qualification gain, focusing on the individual's progression. For example, in a lending scenario, the reward is the profit the bank earns, while the qualification gain is defined as the increase in the individual's credit score. This distinction is crucial because it highlights the trade-off between the utility of the decision-maker and the long-term advancement of the individuals affected by the decisions. By separating these two aspects, our framework can more effectively address systemic inequalities and focus on the intrinsic connection between long-term and short-term fairness.
>
> [1] Wen, Min, Osbert Bastani, and Ufuk Topcu. "Algorithms for fairness in sequential decision making." International Conference on Artificial Intelligence and Statistics. PMLR, 2021.

---

> > ### Comment · Reviewer_hawr · 2024-11-22
> > **Post-Rebuttal Comment**
> >
> > Thanks for the provided update. It addresses my concern about prior work on fair RL.
> >
> > Can you elaborate more about how the scaling can affect the performance? Has the draft been updated to showcase this?

---

> > > ### Author Response · Authors · 2024-11-22
> > >
> > > Thanks for your response. We think the more effective way to elaborate this effect is to plot the bank profit instead of the bank cash, as profit starts at $0 and is independent of the initial cash. We will redraw the plot accordingly and incorporate it into the next version of the draft, which will be submitted later during the rebuttal period.

---

> > > > ### Comment · Reviewer_hawr · 2024-11-22
> > > >
> > > > That would be helpful. Thank you! I am still leaning towards acceptance but I am not an expert in the area.

---

### Official Review · Reviewer_YG39 · 2024-11-04

**Soundness:** 2
**Presentation:** 3
**Contribution:** 2
**Rating:** 5
**Confidence:** 4

**Summary:**

This paper studies long-term fairness through a causal decomposition framework. In particular, the authors first show that the total causal effect can be decomposed into a direct impact, delayed impact, and spurious impact. Moreover, a connection between benefit fairness and the causal effect of switching from a baseline policy to a virtual policy is established. In order to promote long-term fairness, the authors propose to update the PPO objective by introducing a fairness regularizer. Then the experiment section compares various methods on a simulated environment and the proposed method attains improved benefit fairness compared to the other methods.

**Strengths:**

1. The decomposition of long-term fairness into direct and delayed impact is interesting, although it is not surprising. The connection between benefit fairness and the proposed measure of fairness (through policy intervention in an MDP) is quite interesting.

2. The experimental section shows that the proposed modification of the PPO objective works, and in particular it can achieve approximate benefit fairness.

**Weaknesses:**

1. The authors claim that they introduce a general framework to study long-term fairness. However, the proposed framework is based on MDP which has been extensively studied in dynamic fairness.

2. The causal decomposition result is very similar to existing decomposition results of causal fairness for static settings. In fact, the proof is very similar and the only difference seems to be that the indirect effect is replaced with the delayed impact.

3. Finally, the main drawback of the work is that there is no provable guarantee that the proposed method achieves long-term fairness. Since the result is not evaluated across a large class of benchmarks/simulation studies it is difficult to tell whether the method attains long-term fairness in different types of MDPS.

**Questions:**

1. Is there a typo in Proposition 1 (main result)? The inner sum already marginalizes the feature $x'$.

2. You have expressed DPE in terms of benefit fairness. Can the same be done for the delayed impact effect?

---

> ### Author Response · Authors · 2024-11-21
> **W1&2**
>
> Thank you for recognizing that our method aligns with well-accepted methodologies. However, extending causal decomposition to the MDP framework is not trivial. The causal decomposition of the return or advantage function is a non-trivial problem, as highlighted in recent research [1]. In our context, existing methods, either in causal fairness for static settings or in advantage causal decomposition, cannot be directly applied because our objective is to elucidate the intrinsic connection between long-term and short-term fairness and to derive insights that inform algorithm design. To address this gap, we have introduced a novel decomposition of long-term fairness into direct and delayed impacts, which not only advances theoretical understanding but also provides practical guidance for algorithm development.
>
> [1] Pan, Hsiao-Ru, and Bernhard Schölkopf. "Skill or Luck? Return Decomposition via Advantage Functions." The International Conference on Learning Representations (2024).

---

> ### Author Response · Authors · 2024-11-21
> **W3**
>
> While we acknowledge that our proposed method does not include a formal guarantee for achieving long-term fairness, which is largely due to the complexity of analyzing the convergence properties of complex reinforcement learning algorithms like PPO, we believe our work makes significant contributions in other critical areas. Specifically, our research provides a novel perspective by unveiling the intrinsic connection between long-term and short-term fairness in reinforcement learning, a link that has not been thoroughly explored in existing literature.
>
> Regarding the mention of performative prediction by some reviewers, we would like to clarify that performative prediction is a different framework focused on model-dependent distribution shifts in machine learning models. Although it is a powerful approach that can offer theoretical convergence guarantees, it has its limitations. For instance, due to its complexity, it often considers decoupled performative risk, where the data distribution is influenced by parameters different from those being optimized. Compared with performative prediction, we believe that our work also has its own merits in advancing the discussion on long-term fairness.

---

> ### Author Response · Authors · 2024-11-21
> **Q1&2**
>
> Q1
>
> Thanks for pointing out this issue. It might be a slight misuse of notations but not a typo. The two x’ have different meanings. To avoid confusion, we have revised one of the x’ to x’’.
>
> Q2
>
> No, the Indirect Policy Effect (IPE) cannot be expressed in terms of benefit fairness. This is one of the insights gained from our analysis, which shows that the IPE is less sensitive to benefit fairness, and enforcing benefit fairness may not result in a significant change in the IPE.

---

### Author Response · Authors · 2024-11-23
**Revised version uploaded**

Dear Reviewers,

We sincerely appreciate your valuable feedback and have uploaded a revised version of the draft that incorporates your suggestions. The highlighted changes include:
- Revised introduction and related work sections;
- Updated Figures 3 and 4 to include error bars and improve clarity;
- Added Appendix D, which illustrates the influences of $\beta^{KL}$ and $\beta^{\Lambda}$.

Thank you for your time and consideration. We look forward to any further comments you may have.

---

### Meta-Review · Area_Chair_qSYE · 2024-12-20

**Metareview:**

The paper presented a causal framework to characterize long-term fairness in a Markov decision process. The specific fairness is defined  by "qualification gain disparity”.  Using the framework, the authors are able to decompose the cause of unfairness into three types, the direct impact, delayed impact and spurious effects. The paper then develops a PPO-style RL algorithm to enforce the long-term fairness. The causal framework is quite insightful and I believe offers new perspectives to the causes and mitigations of long-term unfairness.

**Additional Comments On Reviewer Discussion:**

The authors were able to address many concerns raised by the reviewers. Overall, I believe the merit of the paper outweighs the weakness.

---

### Decision · Program_Chairs · 2025-01-22

Accept (Poster)